Population genetic structure and variability in Lindera glauca (Lauraceae) indicates low levels of genetic diversity and skewed sex ratios in natural populations in mainland China

Xiong Biao bxiong@gzu.edu.cn xiongbiao8341@hotmail.com 1
Zhang Limei 2
Dong Shubin 2
Zhang Zhixiang zxzhang@bjfu.edu.cn 2
1 College of Tea Science, Guizhou University , Guiyang , China
2 Beijing Forestry University , Beijing , China
Wehenkel Christian
Electronic publication date: 2020 Jan 3
Publication date: 2020
Volume: 8
Electronic Location ID: e8304
Received 2019 Mar 16; Accepted 2019 Nov 27
Copyright: ©2020 Xiong et al.
Copyright year: 2020
Copyright holder: Xiong et al.
License: This is an open access article distributed under the terms of the Creative Commons Attribution License, which permits unrestricted use, distribution, reproduction and adaptation in any medium and for any purpose provided that it is properly attributed. For attribution, the original author(s), title, publication source (PeerJ) and either DOI or URL of the article must be cited.
License URL: https://creativecommons.org/licenses/by/4.0/

Keywords: Lindera glauca, Genetic structure, Apomixis, SSR marker, Gene flow

Funding: Chinese Key Technology Research Development Program of the Twelfth Five-Year Plan 2013BAD01B06 National Natural Science Foundation of China 31900272 Construction Program of Biology First-class Discipline in Guizhou GNYL[2017]009 College and Education committee project of Guizhou University 703/702534183301 This research was supported by the Chinese Key Technology Research and Development Program of the Twelfth Five-Year Plan (No. 2013BAD01B06), the National Natural Science Foundation of China (Grant No. 31900272), the Construction Program of Biology First-class Discipline in Guizhou (GNYL[2017]009) and the college and education committee project of Guizhou University (703/702534183301). The funders had no role in study design, data collection and analysis, decision to publish, or preparation of the manuscript.

==============================
Lindera glauca (Lauraceae) is a tree of economic and ecological significance that reproduces sexually and asexually via apomictic seeds. It is widely distributed in the low-altitude montane forests of East Asia. Despite the potential implications of a mixed reproductive system in terms of genetic diversity, few studies have focused on this aspect. In this study, the genetic structure of wild populations of L. glauca was investigated via genetic analyses. Overall, 13 nuclear microsatellites (nSSRs) and five chloroplast microsatellites (cpSSRs) were used to genotype 300 individual plants, taken from 20 wild populations (a small sample size in some wild populations is due to the limitation of its specific reproduction, leading to certain limitations in the results of this study) and two cultivated populations ranging across nearly the entire natural distribution of mainland China. The populations exhibited low levels of genetic diversity (nSSR: AR = 1.75, Ho = 0.32, He = 0.36; cpSSR: Nb = 2.01, Hrs = 0.40), and no significant effect of isolation by distance between populations existed, regardless of marker type (nSSR: R2 = 0.0401, P = 0.068; cpSSR: R2 = 0.033, P = 0.091). Haplotype networks showed complex relationships among populations, and the H12 haplotype was predominant in most populations. Analyses of molecular variance obtained with nuclear markers (Fsc = 0.293, FST = 0.362) and chloroplast markers (Fsc = 0.299, FST = 0.312) were similar. The migration ratio of pollen flow versus seed flow in this study was negative (r = −1.149). Results suggest that weak barriers of dispersal between populations and/or the similarity of founders shared between neighbors and distant populations are indicative of the gene flow between populations more likely involving seeds. Wild L. glauca in mainland China was inferred to have highly skewed sex ratios with predominant females. In addition, some populations experienced a recent bottleneck effect, especially in Gujianshan, Chongqing, and southwest China (population GJS). It is suggested that few wild male individuals should be conserved in order to maintain overall genetic diversity in the wild populations of this species. These findings provide important information for the sustainable utilization and preservation of the overall genetic diversity of L. glauca.

Introduction

Plant populations respond to the changing environment and climate via phenotype shifts (Nicotra et al., 2010) and, mainly through sexual reproduction, by bringing together high-fitness alleles that reside in different individuals (Whitton et al., 2008). In general, sexual reproduction is predominant in eukaryotes and a nearly universal characteristic of angiosperms. In some groups, however, some sexual plants can reproduce asexually via apomixis, which is the production of clonal seeds in the absence of fertilization (Richards, 1986; Daniel et al., 2001), producing exact genetic replicas of maternal plants (Daniel et al., 2001). Apomixis occurs in fewer than 1% of flowering plant species, with an uneven distribution among lineages (Whitton et al., 2008). Apomixis occurs sporadically among angiosperms (APG, 2003) (Whitton et al., 2008). In some genera (i.e., Taraxacum), apomictic clones can be widely distributed and are temporarily ecologically successful (Van Dijk, 2003; Majesky et al., 2012). However, lack of diversity, the limited possibility of acquiring heritable variability (Richards, 1996), and an increased mutation load leading to the extinction of clones (Van Dijk, 2003), give apomicts an adaptive disadvantage. In contrast, apomicts have lower reproductive costs over than sexuals, a high proportion of loci fixed to heterozygous conditions, and significant advantages over sexuals in colonizing new areas (Majesky et al., 2012). Due to these short-term advantages, natural populations of apomicts are of interest for agricultural development.

There are two major types of apomixis, adventitious embryony and gametophytic apomixis that differ in the way embryos are formed (Whitton et al., 2008; Lo, Stefanovic & Dickinson, 2009). The origin of the former is somatic tissue surrounding the fertilized ovule, and the origin of the latter is an unreduced megagametophyte. Adventitious embryony is widely distributed in nature, and gametophytic apomixis is reported in a few families, e.g., Asteraceae, Poaceae, and Rosaceae. There are two ways for these to spread across space, direct dispersal via apomictic seeds, and indirect transmission via pollen (Whitton et al., 2008). For indirect transmission via pollen, the genes for maternal clonality can be transmitted via male gametes, and this mode of transmission may well be important in the establishment and spread of apomixis (Brock, 2004; Preite et al., 2015). Therefore, the transmission of apomixis genes to sexuals via pollen may be of long-term importance for the spread of apomixis, especially for an agriculturally important tree such as Lindera glauca.

Lindera glauca (Sieb. et Zucc.) Blume (Lauraceae), a deciduous shrub or small tree with both apomixis (asexual reproduction by seeds) (Dupont, 2002) and a sexual reproduction system (Tsui, Xia & Li, 1982; Tsui & Werff, 2008), is distributed extensively in low-altitude montane forests of central and southern mainland China, as well as in Japan, Korea, Vietnam and Taiwan (Wang, 1972; Chang, 1976; Zheng, 1983). As one of the main trees making up the shrubbery and young forest ecosystems in the central and southern areas of mainland China, this species has both economic value and ecological importance. Its fruits are rich in fatty acids and aromatic oils, and they contain terpenoids, flavonoids, and alkaloids, which are used for various applications in traditional medicine. Fruits are also used as raw materials to produce medicines, lubricants, and biochemical products (Zheng, 1983; Wang et al., 1994; Kim et al., 2014; Suh et al., 2015; Qi et al., 2016). Some root extract components, like N-methyllaurotetanine, exhibit significant anti-tumor metastatic activity (Kim et al., 2014; Suh et al., 2015) and some volatile oils from the leaves are used in the industrial production pf spices (Qi et al., 2016). Additionally, the L. glauca species has emerged as a novel potential source of biodiesel in China due to the high quality and quantity of its fruit oil (Lin et al., 2017; Xiong, Dong & Zhang, 2018). There has been increased scientific interest in the species, but relatively little remains known about its reproductive modes and their potential effects on genetic diversity in population dynamics and population differentiation.

L. glauca is native to the mainland China, and diploids (2n = 24) (Yang, 1999) with sexual reproduction and male plants have been known to exist in continental East Asia for several decades (Wang, 1972; Tsui, Xia & Li, 1982; Tsui & Werff, 2008). In a study conducted in Japan, Dupont (2002) found that female L. glauca could asexually reproduce via seeds. Adult population sex ratios of other Lindera species observed in Japan ranged from equal to a strong male bias (including L. obtusiloba, L. umbellata, and L. erythrocarpa) (Dupont, 2002). However, recent empirical studies revealed how L. glauca males are very rare in mainland China, with females reproducing via apomixis. This indicates that natural populations have a mixed reproduction mode that includes apomixis and sexual propagation. Apomixis might play a major role in shaping the genetic structure of the species, by limiting gene flow within populations (Daniel et al., 2001). Interpopulation gene flow in plants is mediated by a combination of pollen and seed dispersal (Ennos, 1994). Some natural populations of apomicts retain residual sexual function as pollen donors and thus have the potential to spread apomixis via male gametes, thereby increasing the genetic diversity observed within apomictic populations (Whitton et al., 2008). In previous studies and records (Wang, 1972; Chang, 1976; Zheng, 1983; Dupont, 2002; Tsui & Werff, 2008), L. glauca is dioecious, and has bisexual or functionally unisexual flowers. However, our survey indicates that there was a very small amount of pollen from the staminode of female flowers (By 2, 3, 5-Triphenyltetrazolium chloride, or TTC method) (Hu, 1993), implying that they have the potential for natural pollination.

Thus, the genetic diversity and structure of natural populations of L. glauca may well be more complex than previously thought. It is essential to study the gene flow and estimate the relative rates of pollen and seed migration among natural populations. Furthermore, population bottleneck effect is thought to be responsible for the very low levels of genetic variation found in a number of species that now have large population sizes (Pannell, 2013). Given that there are very few males of L. glauca in mainland China over the last decade, and many males grown on Taiwan (Zhang, 2007), it is interesting to investigate whether natural populations in mainland China experienced a bottleneck effect or not?

In plants, organelle genomes are often uniparentally inherited (Wills et al., 2005). It is widely believed that plastid genomes are inherited through the maternal parent (Zhang & Sodmergen, 2010). Recently strict maternal inheritance of the plastid was observed in around 82% of the species in two large-scale studies totalling over 500 angiosperms (Corriveau & Coleman, 1988; Zhang, Liu & Sodmergen, 2003; Zhang & Sodmergen, 2010). Strict paternal inheritance is rare, and population size matters as the levels of paternal transmission can be as low as 0.03% in determining the mode of inheritance of the chloroplast genome (Wang et al., 2004). Thus, considering the actual sample size per population, we assume that chloroplast DNA is maternal inheritance for L. glauca.

In the present study, the aims were to (1) investigate the genetic diversity of L. glauca populations in the mainland China, (2) detect genetic variation within and differentiation among natural populations, (3) assess the relative importance of pollen and seeds as agents of gene flow, and (4) determine whether natural populations experienced a decline in size (bottleneck effect). Molecular genetic analyses were performed and individuals in 20 wild populations (and two cultivated populations) of L. glauca were genotyped using 13 nuclear and five chloroplast microsatellite markers developed in the previous work (Xiong et al., 2016; Xiong et al., 2018).

Materials and Methods

Sample collection

During field expeditions carried out from 2013 to 2017, 300 individuals were sampled from 20 wild populations and two cultivated populations, representing nearly the entire natural distribution of L. glauca in mainland China (Table 1; Fig. 1A). Most L. glauca individuals are able to form clones via vegetative reproduction with stolons (Tsui, Xia & Li, 1982), as found in our field survey. In order to avoid the collection of several ramets from the same genet, a single sample was obtained from each cluster of shrubs in close proximity to a main tree, excluding the surrounding young branches growing on the ground. Each sample (individual) in same population was collected at least 10 m apart. In some smaller populations, fewer than 10 plants of putatively nonclonal origin were available. Overall sample sizes varied from 5 to 30 per wild population (Table 1). There was definite apomixis in the individuals of two cultivated populations (based on our survey for three years), and we added it to global analysis as a reference for wild populations. In the field, fresh leaves were immediately dried in silica gel after collection, and preserved at room temperature until DNA extraction.

Table 1 Plant material of Lindera glauca analyzed in the current study.

Sample locations and abbreviations of 20 wild and two cultivated population s used in the main text are listed below.

Population code	Number	Samples accession no.	Longitude (E° )	Latitude (N° )	Elevation (m)	Location	
ATM	10	A14-10	115.8602778	31.2243972	646–834	Tianma, Jinzhai, Anhui	
JGS	30	J13-09	114.0883389	31.8658833	203–317	Jigongshan, Xinyang, Henan	
LDZ	20	L14-04	114.2575139	31.9452917	154–261	Dongzhai, Luoshan, Henan	
SJG	15	S14-10	115.5421083	31.7488389	243–476	Jingangtai, Shangcheng, Henan	
NTB	10	N14-04	113.4231917	32.3289139	241–256	Tongbaishan, Nanyang, Henan	
YTH	10	Y14-04	115.8647444	31.0572889	647–734	Taohuachong, Yingshan, Hubei	
DBS	10	D14-09	115.8391528	31.0087028	834–1,003	Dabieshan, Yingshan, Hubei	
HMF	10	H14-09	113.0076417	28.4467750	224–257	Heimifeng, Wangcheng, Hunan	
TMS	10	T14-09	119.4495389	30.3255389	359–432	Tianmushan, Lin’an, Zhejiang	
SQS	5	S15-08	118.0738639	28.9580278	567–572	Sanqingshan, Yushan, Jiangxi	
LYS	5	LY15-07	118.2822889	32.2834417	134–138	Langyashan, chuzhou, Anhui	
KYS	5	K15-09	121.7357528	37.2661972	136–141	Kunyushan, Muping, Shandong	
FJS	8	F15-05	108.7698111	27.8495556	586–597	Fanjingshan, Tongren, Guizhou	
WYS	7	W16-04	117.9581530	27.6423810	206–217	Wuyishan, Wuyishan, Fujian	
ZJS	30	Z17-04	118.8264001	32.0671503	238–256	Zijinshan, Nanjing, Jiangsu	
WJS	15	WJ17-04	107.4861333	31.2340670	763–791	Wangjiangshan, Dazhou, Sichuan	
GJS	22	G17-04	106.6017170	28.9666002	1,104–1,186	Gujianshan, Qijiang, Chongqing	
NHS	30	HS17-05	112.7194830	27.2638000	233–255	Nanyuehengshan, Hengyang, Hunan	
FHS	8	FH17-04	108.4688137	32.8432889	813–897	Fenghuangshan, Hanyin, Shanxi	
ZJJ	30	ZJ17-05	110.5002670	29.1373330	291–368	Huilongshan, Zhangjiajie, Hunan	
SZYa	5	SZ16-10	121.1781194	31.0778861	45	Shanghai Botanical Garden, Shanghai	
HZYa	5	HZ15-09	113.3594389	23.1793639	11	South China Botanical Garden, Guangzhou, Guangdong	
Notes.

a cultivated population. SZY cultivars were introduced from Jiangsu between 1973–1977. HZY cultivars were introduced from Guangdong in before 1985; all Samples accession numbers refer to voucher specimens deposited in the Beijing Forestry University (BJFU); geographic coordinates and elevation were obtained with portable GPS receiver.

Figure 1 (A) Chloroplast haplotypes and sampling location present in Lindera glauca populations analyzed in the present study (see Table 1 for details). Each population is represented by a triangle, and pie charts are shown when a population was present in more than one haplotype. The green background shows the provincial-level distribution of the species in China. (B) Haplotype network generated with the TCS program. Each haplotype is represented by a single color, and circle sizes correspond to the relative frequency of a particular haplotype in the total sample.

DNA extraction and microsatellite genotyping

Genomic DNA was extracted from 100–150 mg of dried leaves per sample using a modified cetyltrimethylammonium bromide (CTAB) method (Doyle & Doyle, 1987). Microsatellite loci of all 300 individuals of L. glauca were screened, including 13 polymorphic nuclear microsatellite markers (EST-based microsatellites; hereafter nSSRs; Table S1) and five polymorphic chloroplast microsatellite markers (hereafter cpSSRs; Table S2). All nSSRs and cpSSRs were labeled with fluorescently labeled nucleotides (forward primer with M13F) and detected by capillary gel electrophoresis. Subsequent\ steps and the PCR assay were conducted according to Xiong et al. (2016). Genotyping was performed using an ABI 3730XL DNA Analyzer (Applied Biosystems, California, USA ) with a GeneScan 500 LIZ Size Standard, and alleles for each locus were manually scored using GeneMarker version 2.2.0 software (SoftGenetics, State College, PA, USA).

Data analyses

Raw data matrices containing information of alleles and haplotypes for 13 nSSR and 5 cpSSR loci were checked for scoring errors. All SSR analyses were conducted with 300 samples. Data editing and formatting were performed using GenAlEx v. 6.502 (Smouse, Whitehead & Peakall, 2015).

The related indexes of genetic diversity were calculated as followed: for the nSSR data set, genetic diversity indices, including the number of alleles (NA), observed heterozygosity (Ho), expected heterozygosity (He), percentage of polymorphic loci (PPB), Wright’s inbreeding coefficient (FIS), and Nei’s (Nei, 1978) genetic distances, were estimated using GenAlEx v. 6.502 (Smouse, Whitehead & Peakall, 2015) and POPGENE v. 1.32 (Yeh, Yang & Boyle, 1999). The online package GENEPOP v. 4.1.4 (Rousset, 2008) was used to perform exact Hardy–Weinberg equilibrium (HWE) tests and to test for the presence of private (null) alleles. The differentiation index FST was computed for pairs of populations using Arlequin v. 3.5.1.3 (Excoffier, Laval & Schneider, 2005). Allelic richness (AR) was calculated using the software FSTAT v. 1.2 (Goudet, 1995). For the cpSSR data set, number of haplotypes (Nb), genetic diversity (Dv), haplotype richness (Hrs), the number of private alleles (Prv), and the polymorphism information content (PIC) per locus were estimated using HAPLOTYPE v. 1.05 (Eliades & Eliades, 2009).

The population genetic structure was analyzed as followed: for the nSSR data set, the genetic structure of the 22 populations (20 wild and 2 cultivated) was analyzed using the Bayesian clustering approach implemented in STRUCTURE v. 2.3.4 (Pritchard, Stephens & Donnelly, 2000), assuming an admixture model. In order to determine the most appropriate number of genetic clusters or groups (K value), K was set from 1 to 20, and the analysis was run with 20 iterations for each K with a burn-in of 1,000,000 generations followed by 50,000 generations for the Markov chain Monte Carlo (MCMC) simulation. The admixture level for each individual (Q) was also inferred. The program STRUCTURE HARVESTER v. 0.6.94 (Earl & Von Holdt, 2012) was used to estimate the number of population clusters based on the ΔK parameter according to Evanno, Regnaut & Goudet (2005). Based on the most appropriate number of clusters suggested by Bayesian clustering, analysis of molecular variance (AMOVA) was performed using Arlequin, with 10,000 iterations for the permutation test. A neighbor-joining (NJ) tree was generated using POPTREE2 (Takezaki, Nei & Tamura, 2010) based on pairwise Nei (1978) genetic distances between populations determined by GenAlEx. For the cpSSR data set, the Arlequin was used to determine pairwise FST values among all populations. A parsimony network illustrating genetic relationships among haplotypes of L. glauca populations was generated using TCS v.1.1 (Clement, Posada & Crandall, 2000).

Considering that isolation by distance (IBD) can be a key factor keeping populations apart by limiting gene flow (Coyne & Orr, 2004), the IBD of wild L. glauca inter-population in mainland China was tested. In view of the potential importance of pollen in the spread of apomixis (Mogie, 1992; Whitton et al., 2008), the pollen/seed migration ratio (r) was calculated. In order to examine IBD, the Mantel test was performed using GenAlEx, correlating the pairwise genetic distances [FST∕(1 − FST)] with the pairwise geographic distances (in kilometers). To calculate r, we used the followed formula: r = mp∕ms = [(1∕FST(n) − 1)(1 + FIS) − 2(1∕FST(cp) − 1)]∕(1∕FST(cp) − 1) (Ennos, 1994; Petit et al., 2005), where mp is the pollen migration rate, ms is the seed migration rate, FST values (overall FST) are population differentiation estimates derived from AMOVA, FST(n) is the nuclear FST and FST (cp) is the chloroplast FST.

Population bottlenecks were evaluated using BOTTLENECK v. 1.2.02 (Piry, Luikart & Cornuet, 1990) with the infinite alleles model (IAM) that a single mutation is allocated at a time and the resulting number of alleles is computed, stepwise mutation model (SMM) that is a Bayesian approach and generally more appropriate when testing microsatellite loci, and two-phased model (TPM) that is a modified SMM. According to Piry et al., sign tests, Wilcoxon tests, and mode-shift were applied, excluding standardized differences tests, which are useful when at least 20 polymorphic loci are available.

Results

Genetic variation

A total of 74 alleles and 13 haplotypes were identified at 13 nSSRs and 5 cpSSRs across 300 individuals of L. glauca. For each locus, the number of alleles for 13 nSSR loci ranged from 3 (P-298) to 8 (XBLG-060), with a mean of 5.7 alleles (Table S1). In particular, AR ranged from 1.807 to 2.774, with a mean of 2.329, and PIC ranged from 0.363 to 0.711, with a mean of 0.556. Ho and He varied between 0.210 and 0.563, with a mean of 0.380, and between 0.380 and 0.754, with a mean of 0.602.

The total number of alleles of 20 wild populations across 13 nSSR loci varied from 22 (population SZY) to 44 (population ZJJ), with a mean of 32.9, and allelic richness ranged from 1.231 to 2.011, with a mean of 1.740 (Table 2). Population Ho ranged from 0.108 to 0.708 and population He from 0.106 to 0.477, with means of 0.328 and 0.376. The PPB ranged from 53.85% to 100%, with a mean of 85% (Table 2). The values of total number of alleles, Ho, He, and PPB were all smaller in the two cultivation populations, with a mean of 18.5, 0.212, 0.145, and 42.31%, respectively. These parameter values of Ho (0.212), He (0.145), and PPB (42.31%) in two cultivated populations were much lower. Significant deviations from the HWE indicating a heterozygote deficiency were detected in 9 of 22 populations. For some populations (KYS, WYS, ZJS, WJS, GJS, NHS, FHS, and ZJJ), negative FIS values within the populations were observed, indicating more heterozygotes than expected. However, none of the 13 nSSR loci with the heterozygote excesses appear when calculating it on all samples (Table S1), and there is no evidence of private alleles within the data set. Among all wild samples (290), 277 individual plants had a unique multi-locus pattern after PCR amplification with 13 nSSRs primers, indicating that these samples were from different individuals. Of the remaining 13 individual plants, 5 pairs exhibited the same multi-locus pattern in pairs, and 3 individuals exhibited the same multi-locus pattern.

Table 2 Genetic diversity within populations of L. glauca revealed by 13 nSSR and 5 cpSSR markers.

Population	nSSRs	cpSSRs	
	NAnSSR	AR	Ho	He	FIS	PPB (%)	NAcpSSR	Nb	Prv	Hrs	D2sh	
ATM (10)	31	1.823	0.262	0.400	0.3917*	84.62	1	1.000	0	0.000	0.000	
JGS (30)	40	1.837	0.390	0.400	0.043	100	8	4.018	1	0.777	39.638	
LDZ (20)	34	1.689	0.192	0.344	0.4608*	92.31	4	1.527	0	0.363	1.091	
SJG (15)	38	2.011	0.241	0.471	0.5139*	92.31	2	1.142	0	0.133	0.427	
NTB (10)	35	1.916	0.162	0.431	0.6561*	92.31	1	1.000	0	0.000	0.000	
YTH (10)	41	1.992	0.239	0.456	0.5169*	84.62	1	1.000	0	0.000	0.000	
DBS (10)	29	1.588	0.108	0.288	0.6571*	84.62	2	1.220	0	0.200	4.000	
HMF (10)	30	1.636	0.115	0.313	0.6617*	92.31	4	2.381	1	0.644	13.227	
TMS (10)	38	1.987	0.239	0.447	0.5071*	100	1	1.000	0	0.000	0.000	
SQS (5)	26	1.711	0.231	0.317	0.3717*	69.23	3	2.273	0	0.700	3.200	
LYS (5)	29	1.747	0.277	0.319	0.238	61.54	3	2.778	0	0.800	16.480	
KYS (5)	30	1.734	0.415	0.319	−0.200	76.92	2	1.471	0	0.400	0.320	
FJS (8)	28	1.771	0.327	0.380	0.205	92.31	7	6.400	1	0.964	24.343	
WYS (7)	22	1.231	0.264	0.188	−0.333	53.85	2	1.324	1	0.286	8.229	
ZJS (30)	37	1.431	0.708	0.477	−0.470	100	4	1.230	1	0.193	3.915	
WJS (15)	31	1.392	0.518	0.396	−0.275	76.92	3	1.923	0	0.514	2.011	
GJS (22)	28	1.929	0.479	0.382	−0.233	76.92	3	2.142	0	0.558	22.940	
NHS (30)	40	1.823	0.469	0.404	−0.145	100	5	1.531	0	0.359	2.231	
FHS (8)	26	1.748	0.462	0.338	−0.307	69.23	1	1.000	0	0.000	0.000	
ZJJ (30)	44	1.806	0.472	0.453	−0.025	100	9	4.787	1	0.818	7.342	
Mean	32.9	1.740	0.328	0.376	−0.137	85	3.3	2.057	0.3	0.386	7.470	
SZYa (5)	16	1.692	0.192	0.108	−0.667	23.07	2	2.000	1	1.000	0.800	
HZYa (5)	21	1.921	0.231	0.182	−0.059	61.54	1	1.000	0	0.000	0.000	
Mean	18.5	1.807	0.212	0.145	−0.363	42.31	1.5	0.500	0.5	0.500	0.400	
Notes.

a cultivated population.

NanSSR number of alleles across 13 nuclear SSR loci

AR allelic richness

Ho observed heterozygosity

He expected heterozygosity

FIS inbreeding coefficient

PPB percentage of polymorphic loci

NacpSSR number of alleles across five chloroplast SSR loci

Nb number of haplotypes

Hrs haplotype richness

Prv private haplotypes

D2sh the mean genetic distance between individuals

* significant deviations from HWE determined by a global multilocus test implemented in GENEPOP (P < 0.005).

Genetic diversity parameters for cpSSR loci are summarized in Table S2. All 5 cpSSR loci exhibited 2–3 alleles per locus across all samples. Dv ranged from 0.115 to 0.249 per locus, and PIC varied from 0.155 to 0.218. Analyzing combinations of all alleles, there were 22 unique haplotypes (hereafter H; Fig. 1). All populations contained several haplotypes, except for populations ATM, YTH, NTB, FHS, TMS, and HZY (Fig. 1A). The network of plastid haplotypes was complex (Fig. 1B). Haplotype H12 exhibited the highest frequency and was detected in 18 of 20 wild populations (Table S3). Of the 22 haplotypes, 7 were private haplotypes excluding cultivated populations, Hrs per population ranged from 0 (populations ATM, NTB, YTH, TMS, and FHS) to 0.964 (population FJS), and the mean genetic distance between individuals (Dsh2) varied from 0 to 39.638 (population JGS) (Table 2).

Genetic clustering and population differentiation

A Bayesian analysis based on 13 nSSRs implemented in STRUCTURE showed the presence of 2 clusters (K = 2), with only slight admixture at the individual level in each population, except for population ATM (Fig. 2A). The ΔK statistic developed by Evanno et al. indicated that the overall differences were not substantial (Fig. 2B). Cluster orange included 13 wild populations (ATM, JGS, LDZ, SJG, NTB, YTH, DBS, HMF, TMS, SQS, LYS, KYS, and WYS), and the remaining 7 wild populations and 2 cultivated populations were assigned to cluster blue. The NJ tree (Fig. 3A) and principal coordinates analysis (Fig. 3B) based on the nSSR dataset supported the results of STRUCTURE analysis, indicating that 22 populations could be grouped into 2 clusters. However, the network diagram of all 22 unique plastid haplotypes revealed by 5 cpSSRs was complex (Fig. 1B), and failed to support the 2 distinct clusters revealed using nuclear data.

Figure 2 (A) Bayesian inference using STRUCTURE (K = 2) based on 13 nSSR markers from 22 populations of L. galuca, (B) K = 2 appeared to be the optimal number of clusters by showing the ΔK at its peak.

Figure 3 (A) Neighbor-joining (NJ) dendrogram based on Nei’s (1978) genetic distances among populations; (B) principal coordinates analysis (PCoA) of genetic variation across 22 populations of L. glauca based on 13 nSSR markers.

The 2 clusters revealed by STRUCTURE analysis were set as groups for AMOVA based on both the nSSR and cpSSR dataset (Table 3). Using the nSSR dataset, the majority of genetic variation was detected within populations (63.82%), indicating a genetic differentiation mostly at the individual level. Nevertheless, a considerable proportion of the total variation (26.42%) was found among populations within groups, and a small amount of variation (9.76%) occurred among the 2 groups. In contrast to the nSSR results, the AMOVA based on the cpSSR dataset showed that a larger proportion of genetic variation could be attributed to variation within populations (68.84%) and among populations within groups (29.37%), and little variation among groups (1.79%). The overall FST values calculated by AMOVA were 0.362 (P ≤ 0.0001) for the nSSR dataset and 0.312 (P ≤ 0.0001) for the cpSSR dataset.

Table 3 Analysis of molecular variance (AMOVA) and degrees of freedom (df) based on 13 nuclear SSR and five chloroplast SSR markers for populations of L. glauca.

The groups revealed by a Bayesian STRUCTURE analysis (K = 2) were considered for both marker types.

Source of variation	nSSRs	cpSSRs	
	df	% of variation	F-statistics	df	% of variation	F-statistics	
Among groups	1	9.76	FCT = 0.09756*	1	1.79	FCT = 0.01793	
Among populations within groups	20	26.42	FSC = 0.29281*	20	29.37	FSC = 0.29903*	
Within populations	273	63.82	FST = 0.36180*	273	68.84	FST = 0.31160*	
Notes.

FCT differentiation among groups

FSC differentiation among populations within groups

FST differentiation among populations

* Significant values with P ≤ 0.0001.

Isolation by distance and pollen/seed migration ratios

The estimates of genetic differentiation (FST value) based on 13 nSSRs ranged from 0.023 (between WJS and GJS) to 0.427 (between LDZ and FHS) (Table S4), excluding 2 cultivated populations. Only 4 pairwise comparisons (ATM and JGS, WJS and NHS, NTB and YTH, and WJS and GJS) showed significant Fst values (P ≤ 0.05). Adopting a P-value of 0.01, no significant correlation between pairwise genetic distance [FST/(1 − FST)] and geographic distance (in kilometers) was found using the Mantel test (Fig. 4) for the nSSR dataset (R2 = 0.0401, P = 0.068) or the cpSSR dataset (R2 = 0.033, P = 0.091), suggesting that L. glauca in mainland China does not exhibit significant IBD. The FST values for the nSSR and cpSSR were similar (Table 3), and the r was −1.149, indicating that most gene flow among populations occurs via seed, rather than via pollen.

Figure 4 (A) Figure plot of geographical distance against genetic distance for 22 populations of L. glauca based on 13 nSSR markers (B) and five cpSSR markers.

Population bottleneck effect

Several populations had a significant excess of heterozygosity expected at mutation-drift equilibrium (i.e.,  He > Heq) (Piry, Luikart & Cornuet, 1990) under the 3 models in the bottleneck analysis, which indicated a deviation from mutation drift equilibrium in wild L. glauca populations (Table 4). More specifically, population GJS exhibited a significant bottleneck event according to the sign and Wilcoxon tests in all 3 models, indicating a population size decline (bottleneck effect) in its history. Population WJS experienced a significant bottleneck event according to the sign test and Wilcoxon test for the IAM and TPM methods, and FHS exhibited a significant bottleneck event by Wilcoxon test in all 3 models. SJG, FJS, and ZJS only exhibited a significant bottleneck event based on the sign and Wilcoxon tests for the IAM method (Table 4).

Table 4 Bottleneck analyses for 20 wild populations of L. glauca.

Population	IAM	TPM	SMM	
	Sign test	Wilcoxon test	Sign test	Wilcoxon test	Sign test	Wilcoxon test	
ATM	0.0366*	0.0674	0.0596	0.1230	0.2577	0.2061	
JGS	0.3046	0.2439	0.3968	1.0000	0.4540	0.7354	
LDZ	0.3681	0.3804	0.5193	0.9097	0.4268	0.4697	
SJG	0.0047**	0.0105*	0.0469*	0.0522	0.1750	0.1514	
NTB	0.2277	0.0923	0.2999	0.1514	0.3827	0.6772	
YTH	0.0782	0.2402	0.3010	0.8984	0.2911	0.4131	
DBS	0.5698	0.9658	0.4498	0.7002	0.3485	0.3652	
HMF	0.3258	0.9697	0.4538	0.9097	0.4342	0.5693	
TMS	0.3221	0.2734	0.5841	0.5879	0.0534	0.8394	
SQS	0.1909	0.0371*	0.3072	0.3594	0.2811	0.7344	
LYS	0.5500	0.4609	0.3552	0.8438	0.2784	0.4609	
KYS	0.6099	0.8457	0.1162	0.4316	0.1274	0.1934	
FJS	0.0234*	0.0134*	0.1536	0.0923	0.4000	0.2661	
WYS	0.4625	0.9375	0.3955	0.9375	0.3416	0.5781	
ZJS	0.0117*	0.0067**	0.0247*	0.0803	0.1177	0.2439	
WJS	0.0105*	0.0049**	0.0208*	0.0137*	0.1171	0.1309	
GJS	0.0066**	0.002**	0.0106*	0.0098**	0.0189*	0.0098*	
NHS	0.1557	0.6355	0.2295	1.0000	0.5266	0.7869	
FHS	0.1297	0.0098**	0.1693	0.0098**	0.1244	0.0371*	
ZJJ	0.3329	0.1909	0.3724	0.5417	0.3253	0.5417	
Notes.

* Significant values with P ≤ 0.05.

** Significant values with P ≤ 0.01.

Discussion

Genetic variation within populations

In this study, sampling covered a large portion of the natural distribution, and overall genetic diversity across wild L. glauca populations exhibited low levels based on both nSSR (mean AR = 1.74, Ho = 0.33, He = 0.38, FIS =  − 0.14) and cpSSR (mean Nb = 2.06, Hrs = 0.39) loci. Our estimates of genetic diversity in L. glauca were almost half those of long-lived perennials (Ho = 0.63, He = 0.68), out-crossing species (Ho = 0.63, He = 0.65), and plants with wide distributions (Ho = 0.57, He = 0.62) (Nybom, 2004), which were lower than Laurus nobilis in Lauraceae (AR = 3.22, He = 0.56) (Marzouki et al., 2009). There are several major factors influencing variation that can result, each by itself or in combination, in the low levels of genetic variation observed in wild L. glauca populations. Asexual reproduction through apomictic seeds can decrease genetic variation in a population especially in apomictic populations (Lo, Stefanovic & Dickinson, 2009). Similarly, effective population size could seem like a limitation, due to the population being established from a limited number of individuals and a small sampling quantity. For two cultivated populations, these values were much lower than in wild populations, perhaps because apomixis reduces genetic diversity, or because of the potential confounding effect of small population sample size. Furthermore, the estimated values of genetic diversity therein are also lower than nSSR-based values found in literature (AR = 2.61, He = 0.44, FIS =  − 0.37) (Zhu et al., 2016). Differences in genetic variation between this study and the previous one are likely to be explained by the number of sampling populations and individuals the study of Zhu et al. (2016) included 6 populations, a total of 96 individual plants, while this study included 20 wild populations, making a total of 290 individual plants.

On the other hand, a marked similarity in the molecular variance revealed by the 2 types of markers (overall Fsc = 0.293 and FST = 0.362 for nSSRs; 0.299 and 0.312 for cpSSRs) was observed, indicating general consistency between chloroplasts and nuclear DNA. More specifically, predominant apomixis in wild L. glauca could explain how FST observed for nuclear markers is a little higher than that for chloroplast markers, and vice versa. In addition, apomictic reproduction of L. glauca could also affect the results. Therefore the use of clone-corrected data (removing the data of clones from the same parent) is necessary if clones are detected, because L. glauca trees could form clones via vegetative reproduction with stolons.

Nine populations had more heterozygotes than expected. STRUCTURE analysis showed that the 9 populations were grouped into one group and the rest into another, suggesting negative FIS is an important factor affecting group of population difference. Usually, positive FIS values within populations (JGS, LYS, and FJS) indicated inbreeding, and negative FIS of wild populations (KYS, WYS, ZJS, WJS, GJS, NHS, FHS, and ZJJ) suggested outbreeding. However, considering that clonality probably generates significant negative FIS in some wild plant populations with asexual reproduction when considering all individuals (Stoeckel et al., 2006), the observed negative FIS of wild populations (KYS, WYS, ZJS, WJS, GJS, NHS, FHS, and ZJJ), coupled with the result grouped by NJ tree and the cluster pattern of the NJ tree (blue cluster) (Fig. 3A), suggest that apomixis may have been common in these populations. On the other hand, global multilocus tests indicated that sexual reproduction did exist in these populations that were positive FIS values (0 < FIS < 1). There is mixed reproductive system existed in this species, thus explaining the FIS pattern.

Genetic differentiation among populations

Populations can cluster according to habitat type or geographic distance. However, for species with predominantly asexual populations, like Daktulosphaira vitifoliae (Vorwerk & Forneck, 2006), Crataegus douglasii (Lo, Stefanovic & Dickinson, 2009), and Taraxacum officinale (Majesky et al., 2012), no significant correlation exists between genetic distances and geographic distances. In this study, a correlation between genetic distances (as measured by [FST∕(1 − FST)] values) and geographic distances (in kilometers) was not detected in L. glauca populations, regardless of marker type, suggesting that weak barriers to dispersal between populations and/or the common founders between neighbors, distant populations, and apomictic populations did not completely limit gene flow. Results indicate that sexual dispersion and apomixis co-occurred in the same natural population. However, further research is needed to investigate the extent to which apomixis limits gene flow, and the exact rate of sexual production and apomixis that occurred among and within populations.

According to Ennos’ formula (1994), the migration ratio of pollen flow versus seed flow (r) in this study was negative (−1.149), suggesting that gene flow between populations is more likely to involve seeds. Usually, the value of r is positive in almost all angiosperm species (Ennos, 1994). The biased r value could be explained by the reproductive mode in different populations being the apomictic seeds of females, coupled with the result that no staminate flower were found at field observation sites for 5 consecutive years. Considering that a situation where a few individuals can self-pollinate is likely, pollen from even just a few staminodes of female flowers may be associated with the negative r value of pollen flow versus seed flow. However, because the formula is derived for a hermaphrodite species, it needs to be modified to account for the disproportionate maternal contribution from the females to the next generation if the exact r value of L. glauca can be expected to be observed.

Source of evolutionary potential of apomicts

Although 22 haplotypes were observed across 20 wild populations and 2 cultivated populations of L. glauca, H12 accounted for 62.03% of overall haplotypes and was detected in all populations (Table S3), except for FHS, LYS, SZY, and HZY. This haplotype existing in many populations separated by considerable geographical distances (e.g., greater than 1,720 km between KYS and GJS), coupled with the complex network (Fig. 1B), suggested three inferences to account for the observed result. First, these individuals from different populations had a relatively recent shared female founder. Second, there was an apomictic lineage for this species within some populations. Third, the genome of this species has genetic traits that lead to apomixis, which might be induced by some factors (e.g., biological stimulation, environmental influence, climate changing, etc.), and coexists with sexual reproduction in identical individual plants.

The first inference could be explained by a hypothesis that the H12 haplotype may be associated with the migratory patterns of some birds responsible for the dispersion of apomictic seeds over a long distance. However, given that H12 is the most frequent and connected haplotype, probably is ancestral, this inference is not reliable. Besides, despite conducting field observations for 5 consecutive years, few birds were found eating grown fruits of L. glauca, as well as few small mammals (e.g., Paguma larvata). The second inference means having an early maternal ancestor through apomictic reproduction for many individuals. However, all the living species are at the tips of the tree of life, and apomixis is a derived condition (APG, 2003; APG, 2016; Horandl, 2006; Thompson & Ritland, 2006; Lo, Stefanovic & Dickinson, 2009). Consequently, having ancestral asexual angiosperms is almost impossible, because asexuals fail to maintain sex and recombination in populations that are limited in size, therefore are unable to bring together high-fitness alleles that reside in different individuals. The third inference means that apomicts of L. glauca may be of very recent origin and have the ability to apomictically reproduce through mutations or losses of some sexual genes. A similar situation exists in some species, such as Taraxacum officinale that exhibit alternations of asexual and sexual histories of apomicts (Majesky et al., 2012), some hawthorns (Crataegus; Rosaceae) that have a population genetic structure of diploid sexual and polyploid apomicts (Lo, Stefanovic & Dickinson, 2009), and a marbled crayfish (Procambarus virginalis) that reproduces through parthenogenesis (Gerhard et al., 2003; Ewen, 2018). Therefore, when coupled with the results of filed surveys that show apomixis occurred in all sampled populations, this haplotype (H12) is probably a predominant genotype existing in populations that have the ability to apomixis.

Above all, the authors reject the former two inferences that most populations across such long geographical distances have a relatively recent founder or an ancient apomictic ancestor, and believe the last inference that apomixis caused by mutations or losses of related genes makes this maternal haplotype (H12) present in many individuals from different populations more likely.

Conjecture about histories of natural populations

According to the relationship Ne = 4(NmNf)/(Nm + Nf), where Nm represents the number of males, and Nf represents the number of females (Beerli & Palczewski, 2010), the accurate values of Ne could not be calculated because there were no males in the samples collected. Even so, based on the rarity of male individuals in mainland China, dioecious reproduction reported in the past several decades (Tsui, Xia & Li, 1982; Wang, 1972), some specimens of branches of male individuals stored in the China National Herbarium (PE), and many males grown on Taiwan (Zhang, 2007), it is inferred that some natural populations of L. glauca recently experienced a severe bottleneck and male individuals experienced a decline, most likely resulting from anthropogenic causes. However, in order to test the above hypotheses and obtain more accurate results, further field investigations are necessary, including samples from male and female individuals in a population (especially males found on Taiwan), correcting for clone and apomictic reproduction, across the full range of L. glauca, including Japan, South Korea, and Taiwan.

Implications for conservation

Genetic diversity is recognized as an important population attribute for both conservation and evolutionary purposes (Cena et al., 2006). The purpose of the conservation of endangered and threatened species is to maintain their contributions to overall genetic diversity. People usually focus only on endangered species and provide protection, whereas some species that are reduced in genetic diversity also require protection. This study detected a lower level of genetic diversity in L. glauca than that of some other species in Lauraceae. The conclusion is that wild L. glauca populations have female-skewed sex ratios, which was consistent with our field survey and sampling for 5 consecutive years. The destruction of habitats in order to plant other commercial or medicinal crops and felling by local farmers may explain the low frequency of male L. glauca in mainland China, although the species is common and widely distributed in other regions. The authors propose finding male individuals and promoting sexual reproduction in order to maintain this species’ overall genetic diversity.

Conclusion

This study has shown low levels of genetic diversity in L. glauca across nearly its the entire natural distribution in mainland China. A complex correlation between populations was revealed by haplotype networks. Genetic structure within and among populations was similar at the nuclear and chloroplast levels. Furthermore, some populations experienced a recent bottleneck, and gene flow between populations is more likely to involve seed. This implies that wild L. glauca in mainland China has highly skewed sex ratios with predominant females.

Supplemental Information

Table S1 Genetic characteristics of 13 nuclear SSR loci in 22 populations of Lindera glauca (N= 300)

Click here for additional data file.

Table S2 Genetic characteristics of five chloroplast SSR maerkers and results of genotyping in L. glauca (N= 300)

Click here for additional data file.

Table S3 List of haplotypes detected at five cpSSR loci in 22 populations of L. glauca. Private haplotypes are highlighted in red, and their corresponding frequencies are shown in the last column

Click here for additional data file.

Table S4 Pairwise geographic distances (km) above the diagonal and pairwise FST below the diagonal for 22 populations of L. glauca

Click here for additional data file.

The authors thank Richard A. Ennos for his explanation of the formula of pollen/seed migration ratios. The authors thank , Alejandra Lobo and Jean Beaulieu and one anonymous reviewer for their constructive suggestions about the original manuscript. The authors thank Rodrigo C. Gonçalves-Oliveira for his assistance with the AMOVA analysis, and Jigongshan National Nature Reserve for a part of the material collection.

Additional Information and Declarations

Competing Interests

Author Contributions

Data Availability

The authors declare there are no competing interests.

Biao Xiong conceived and designed the experiments, performed the experiments, analyzed the data, contributed reagents/materials/analysis tools, prepared figures and/or tables, authored or reviewed drafts of the paper, approved the final draft.

Limei Zhang analyzed the data, prepared figures and/or tables, approved the final draft.

Shubin Dong performed the experiments, contributed reagents/materials/analysis tools, prepared figures and/or tables, approved the final draft.

Zhixiang Zhang conceived and designed the experiments, authored or reviewed drafts of the paper, approved the final draft.

The following information was supplied regarding data availability:

Raw data is available in the Supplemental Tables.

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
