# Peer review of "Population genetic structure and variability in Lindera glauca (Lauraceae) indicates low levels of genetic diversity and skewed sex ratios in natural populations in mainland China"

_PeerJ, doi:10.7717/peerj.8304_

## Round 0.1 · original submission · Major Revisions

Reviewer 1 stated that either the introduction or the discussion should also provide citations of other studies that have investigated correlations between reproductive strategy and climatic conditions. Besides, the introduction is very restrictive to the species or at best the genera investigated here. Reviewer 2 noted that some aspects could be reviewed to improve the quality of the manuscript and the interpretation of the results. Please make sure that your manuscript is diligently corrected for language.

[]

Reviewer 1 ·

Basic reporting

Xiong et al. use a tree species with mixed mating system to explore the influence of mating system on levels of genetic diversity. By using a combination of Chloroplast and nuclear microsatellites they reveal overall low genetic diversity and skewed sex ratio matching the pattern detected in Japan. Overall, I am satisfied with the main conclusions but have some major suggestions for the representation of the methods, the motivation behind doing some analyses and the interpretation of results.

The introduction is very restrictive to the species or at best the genera investigated here. Implications for genetic diversity and population structure across other apomictic species and species with mixed mating systems, such as Taraxcum (Majesky et al. 2012), Capsella (Wright et al), Arabidopsis, Plantego and Silene (McCauley 1997) would be useful to broader audience. Either the introduction or the discussion should also provide citations of other studies that have investigated correlations between reproductive strategy and climatic conditions (e.g: Preite et al. 2015; Van Dijk 2003). For instance, why is selfing/apomixis favoured in certain populations despite it causing reduction in genetic diversity.

How does the pattern of diversity they found and the sex ratio relate to the study in Japan on the same species that was also apomictic.

Line 284 to 286: The authors indicate that the two types of markers provided similar results, but in the next sentence they refer to higher differentiation in the nuclear markers. These sentences are contradictory to each other. Please clarify and rewrite.

Highlight the relationship between FIS values, sex ratio skewness and genetic diversity. Do pops with more negative FIS tend to have low genetic diversity.

Highlight the comparison between apomictic and non-apomictic populations, since that is the primary purpose of this work.

Experimental design

I am a little confused on why pollen and seed migration rates were used, specifically because all the pops examined here had no males?

I like the idea of using several methods to detect bottleneck but these approaches need to be justified. For instance, how are they different from each other. A bigger problem here is why did the authors want to access bottleneck to begin with. No-where in the introduction or the methods do the authors reference anything about the demographic history of the species, so the estimation of bottleneck comes out of left field. I would suggest the authors to elaborate on why understanding past demographic history would be useful.

The authors also use several different mutation models in their estimation of Ne, the bearing of these on their inference of bottleneck needs to be explained in the discussion.


The estimation of “r” relies of different modes of inheritance for chloroplast and nuclear markers. This needs to be mentioned somewhere either in the introduction or just before “r” is estimated in methods.

How were the mutation rates for estimating Ne obtained?

Validity of the findings

In line 175-176 & 238 the authors mention about multi-locus pattern and mutli-locus test. This is very ambiguous.

Please clarify what “robust maternal lineage” means and how the authors concluded that it is different from the haplotype found in Japan. In the absence of an analysis done across the two countries, I don’t think the statement listed on line 251-252 is valid and neither does it add any value to the primary purpose of the paper.

It is not clear what the authors are trying to convey via line 261 to 263. Please add more details and reword this sentence.

Additional comments

Line 67: Please indicate year after Dupont and change “reproducing” to “reproduce”.
Line 69: Change “equality” to “equal”
Figure 1: If the green indicates the geographical range of the species this should be mentioned in the figure caption or in the legend.
Line 138 is redundant with line 135.
Line 142 to 144: This sentence is written awkwardly. I would suggest rewording it to emphasise that IBD and pollen/seed migration ratio were assessed to determine whether gene flow was limited between populations.
Line 236 & 240: Change “exited” to “existed”
Line 280: change “pollen that was vitality” to “pollen that was vital”
Table 3: Please complete the caption sentence “were considered for”
Figure 1: Please complete the caption sentence

·

Basic reporting

This manuscript presents relevant analyses to describe the effect of asexual reproduction through apomixis on structure, differentiation and genetic diversity of a perennial bush throughout a partial distribution range, based on the variation of nuclear and organelle genetic markers. Sampling is appropriate for the study and the applied methods are useful to answer the questions posed by the researchers. However, some aspects could be reviewed to improve the quality of the manuscript and the interpretation of the results. As well, most of the sections are written in an understandable way however I detected some errors and some paragraphs are susceptible to be improved. Tables and figures are presented properly, although it is suggested to the authors to look for a public database to present the raw data


I find that the manuscript is based on two assumptions that should be reconsidered or discussed more broadly. The first assumption is that the determination of the sex of the individuals of this species has exclusively a genetic basis. A contradiction arises when the plant is described as dioecious, even though both in the introduction and discussion sections it is mentioned that according to the observations in the field the population seems to present gynomonoecious individuals. A gynomonoecious system suggests absence of sex chromosomes or incipient chromosomal differentiation. Therefore, it is possible that the formation of pistillated or staminated flowers is determined by environmental factors This must be considered both in the interpretation of the data in the discussion section and in the conclusions and implications for conservation. The second assumption is related to the mention that in years previous to the study, the presence of staminate flowers has been reported. Further considering that the species under study is a perennial shrub, it should not be assumed that the absence of trees with staminated flowers in a period of tens of years will have a detectable effect on the genetic variation and the structure of the current populations. This means that the results presented can also be interpreted as a result of ancestral genetic flow.

Abstract:
r.13-14. … a tree of economic and ecological significance with a mixed reproductive system with sexual and asexual reproduction by means of apomictic seeds.
r. 17 … Sex ratio analysis is just mentioned in the text but is not directly analyzed in this work.
r. 24 … Estimates of structure obtained with nuclear and chloroplast markers were similar

Introduction:
The introduction should focus mainly on the possible effects of clonality and particularly apomixis, on the genetic structure and diversity of plant populations.
It would be important to mention in more detail the mechanisms involved in the movement of pollen and seeds.

Experimental design

Materials and Methods and Results sections
Please give more information about cultivated samples, i. e. if there some information about the origin of these artificial populations (r. 94-95).

Give more information about the inference of the parsimony network for chloroplast haplotypes (r. 140-141). Considering that this network is not resolved maybe the analysis could be improved. Moreover, if the options for data were the appropriate to infer this network, maybe authors could resolve some loops using the criteria suggested in Crandall and Templeton (1993; https://www.genetics.org/content/genetics/134/3/959.full.pdf).

r. 155-157 It is recommendable to use 2Nu to estimate Ne for haploid markers. Authors should mention why this mutation rate was assumed for both type of markers.

It is convenient that the authors do not interpret the values of the fixation indices as levels of differentiation. This applies to the results as well as to the discussion and the table 3. For more information you can review Jost et al 2018 (https://onlinelibrary.wiley.com/doi/full/10.1111/eva.12590)

Validity of the findings

Suggestions for discussion section

r.239-246. Negative Fis values are commonly interpreted as the individuals in the population are less related than expected under a model of random mating. Therefore, the conclusion that Fis negative values are result of apomictic reproduction is not fully supported. Maybe authors could find more information in other systems to support this statement.
r. 246-249 The most common haplotype is commonly interpreted as the ancestral one, therefore this interpretation should be reconsidered. What is a “robust maternal lineage”

It is notable that populations of green cluster tend to have greater observed heterozygosis than expected (although according to their analysis they do not show deviations from the HWE), can the authors discuss this?

Please consider these questions as well
The two groups detected in STRUCTURE are related with a geographical pattern?
considering that greater movement of seeds was detected, the divergence between the groups can be associated with the migratory patterns of the birds responsible for the dispersion of seeds?

---

## Round 0.2 · Major Revisions

Please carefully consider all the comments of reviewer 2.

·

Basic reporting

This is a version of the article where the theoretical framework and interpretation of the results show an improvement, however this version shows more flaws in written English.
Although interpretations have been redirected to contrast the effect of sexual reproduction through pollen dispersion with asexual reproduction through apomixis, the assumption about the existence of a male and female genetic lineages, without an evidence for a genetic system of determination of sex in this species, causes a bias in the interpretation of the results.

Experimental design

Authors accepted most of the suggestions and made significant improvements in the article, however it is not yet clear how the effective size of the populations was estimated, as it is unlikely that the effective population size for cytoplastic markers is higher than for nuclear ones. For this purpose, it is necessary to clarify whether the authors show values of the scaled effective population size (theta: 4Nu and 2Nu) or whether they are directly calculating the effective population size (Ne).
According to table 2 It should be noted that populations with larger effective sizes for chloroplast show a single haplotype, which may indicate that there is a failure in the calculation of effective population size.

Validity of the findings

Although the results of the article are valuable, the effective population sizes obtained should be reviewed, so the overall interpretation of the patterns found may need to be remade.

---

## Round 0.3 · Major Revisions

The two reviewers have already checked the manuscript several times. Nevertheless, the quality of the manuscript has not improved significantly and many comments have not been addressed. The authors should therefore prepare their manuscript very carefully to avoid rejection in the next round.

Reviewer 1 ·

Basic reporting

There are still several places where the sentences needs to be re-written. These include: 297, 316 & 357.

I am happy with the incorporation of more context in the introduction and the discussion.

Experimental design

Several of the points I had noted under the experimental design are yet to be addressed. Only in the discussion do the authors bring up issues related with using the equation for "r" and the various ways of interpreting it based on the reproduction mode.

The other points have not been addressed.

Validity of the findings

No comment

·

Basic reporting

I still think that the authors assume, without evidence, a genetic system of sex determination, which biases the approach of the article and the interpretation of the results. In spermatophytes the dominant phase is the sporophyte, while the gametophyte is a microscopic tissue contained in the flower or pollen. Sex chromosomes have been found only in less than 1% of the plants studied so far. Moreover, in different plants has been proven that production of staminate or pistillate flowers could be related to environmental conditions.Therefore, it is advisable to avoid calling individuals as males or females, since probably the sporophyte is not sexually determined and not assume that there is a male or female lineage. Examples of this are found through the document, review particularly rows: 81-83, 104, 125, 303, 315, 377. This assumption is taken to the extreme in results interpretation by assuming that the most abundant haplotype with the most connections in the haplotype network is linked to the recent apomixis (rows 333-334), and not to historical demographic patterns.


The reason why I worry about estimates of effective population sizes, is that under the drift-mutation model it is expected that populations with greater genetic diversity (He or number of haplotypes) have the largest effective sizes, which is not satisfied in these results. I should maintain that a review of the analysis is desirable.



I must insist that written English is deficient, and the document requires a professional review, The following are examples of phrases that need to be reviewed:

r. 17 “a package genetic analysis” should say “a package of genetic analyses”
r. 56 “Adventitious embryony ARE widely distributed in nature, and gametophytic apomixis ARE reported in few families” should say “Adventitious embryony is widely distributed in nature, and gametophytic apomixis is reported in few families”
r. 105 “whether” is used when two alternatives are exposed, for example “whether natural populations experienced a bottleneck or not”
r. 247 “Detaily”
r. 264 “Another explanation is establishment”, should say “Another explanation is that” or “Another explanation could be that”

Moreover, some sentences must be revised, as they seem incomplete or without sufficient support:
r. 38 “The genetic basis for these abilities is sex”… asexual organisms can not adapt?,
r. 39 sexual reproduction is predominant only in eukaryotes
r. 106-108 This is not tested in this work, and natural selection as well could reduce effective population size
r. 323-324 “According to known research… “ , this sentence must say what is according to known research

Experimental design

no comment

Validity of the findings

no comment

Additional comments

no comment

---

## Round 0.4 · Minor Revisions

Please, make the minor revisions which are required by the reviewer.

Reviewer 1 ·

Basic reporting

I don't have much to add with regard to major comments, just a couple more editorial changes that need to be incorporated are listed below:

Line 14: add is after "it"
Line 28 : Please change "In additional, and some" to "In addition, some populations"
Line 81 to 82 need to be rewritten: remove "few" in line 81 and change to "limiting gene flow" and make sure the parenthesis and capitalisation is in the correct place.
Line 109: Should be removed because it is misleading to the readers as to why pollen and seed gene flow was calculated. Please remove or move to the discussion where the presence of vital pollen is mentioned from female flowers.
Line 149: Change to "Considering that isolation by distance (IBD) can be a key factor keeping populations apart by limiting gene flow..."
Line 239: Change "limited" to "limitation".
Line 250: Please elaborate what "clone corrected data" means and how this could alter any results presented here.
Line 247: Please complete "involve seed instead..."

It is still not clear what the authors mean by" robust maternal lineage", this is mentioned in line 308-310. I suggest either moving it up to line 288.

Experimental design

None

Validity of the findings

None

Additional comments

None

---

## Round 0.5 · Minor Revisions

It was found numerous typographical errors, grammatically incorrect sentences and unclear concepts.

The reviewer suggested to make an exhaustive and objective review of the text again.

·

Basic reporting

Although the manuscript has improved in this latest version I have found numerous typographical errors, grammatically incorrect sentences and unclear concepts.
I suggest to make an exhaustive and objective review of the text again.

Authors mention that the species has a dioecious reproductive system, but that there are bisexual flowers with viable pollen therefore it could be an androdioecious reproductive system. I have questioned the authors on different occasions about evidence regarding what mechanism of sex determination they assume, however I note that they make no mention of it. This assumption influences the interpretation of the results and must be supported. Dioecy and sex ratio are commonly influenced by environmental and population factors, if this is the case, it is incorrect to assume that there is a male lineage and a female one, since probably under different conditions the same genotype can produce staminate or pistillate flowers. In the same way it would be useful for the discussion, to mention the evidence that exists regarding the genetic basis of apomixis.

In the attached file you will find some observations over the document. I suggest an exhaustive and objective review of the text again.

Experimental design

No comment

Validity of the findings

No comment

---

## Round 0.6 · Minor Revisions

Numerous typographical errors, grammatically incorrect sentences and unclear concepts were found.

The reviewer suggested to make an exhaustive and objective review of the text again.

·

Basic reporting

no comment

Experimental design

no comment

Validity of the findings

no comment

Additional comments

In this version of the manuscript I still find multiple writing errors, many of which I think had already been pointed out in previous versions. I have reviewed this article multiple times and I consider that the quality of writing and understanding of concepts is not enough for publication. However, I do not want to affect the authors and I would like them to reconsider the option of having the manuscript reviewed by an editorial service. Attached you will find the PDF with observations made to this version.

---

## Round 0.7 · Major Revisions

I have tolerated the many rounds of review because I and the reviewers find the manuscript interesting and worth publishing in PeerJ. But before it can be Accepted, all mistakes must be eliminated. Therefore, all comments are to be understood as assistance to create a correct and attractive article. The 3rd reviewer was brought in because Reviewer 2 had given up on the task after so many rounds. But, I have also found that some problems have been overlooked by the reviewers 1 and 2.

Please consider the comments of the reviewer. Also change the following:
1. lines 202-203: This sentence is part of the discussion.
2. lines 325-328: Check this text; First, "associated with migratory patterns of some birds", then "not associated with bird dispersal patterns".

·

Basic reporting

The study " Population genetic structure and variability in Lindera glauca (Lauraceae) indicates low levels of genetic diversity and skewed sex ratios in natural populations in mainland China", by Xiong, Zhang, Dong, and Zhang, provides an insight into genetic diversity and population structure of Lindera glauca, a tree species growing in China. The species reproduces sexually and asexually via apomictic seeds. This study is adding new knowledge that may assist in sustainable utilization of the species and in the conservation of its genetic resources.
The manuscript is globally well written, although it will require some editorial work. The introduction shows well the context of the study. However, some additional background information should be provided. For instance, I think it would be important to indicate that in angiosperms maternal inheritance is assumed for chloroplast DNA. I do not think that it has been tested for Lindera glauca and the authors take for granted that all the readers know that maternal inheritance is assumed. On line 85, the authors state that dioecious L. glauca has bisexual or functionally unisexual flowers. Dioecious species have distinct male and female individual. Thus, I am not sure that strict dioecy applies to that species otherwise more background information would be welcome.
The structure of the article is fine and tables as well as figures are well designed and of good quality.

Experimental design

The objectives of the study are clearly stated and define the research question. However, I have some concerns that I describe below.
The importance of this work does not seem to me to be well reflected in the sampling design. For instance, the sample size is quite small in more than half of the populations; below the standards of population genetics analyses. The initial average sample size is 13.6 trees/population. Some populations end up with extremely low sample sizes (such as n=10, 8, 5), rendering the corresponding results hard to interpret due to the potential confounding effect of small sample size in the observed genetic parameter values.
Thus, it is not possible to determine if the small sampling size of many populations was because the number of individual making up these natural populations was very small or because the authors did not have the resources to collect more samples, or have access to other larger populations. Thus, more details would be required including the average distance between sampled individuals. There is no background information on the natural population size.
I also wonder why samples of two cultivated populations were included in this study. No arguments are given by the authors about the importance to include these populations in their study. In the aims of the study, they indicate they want to detect genetic variation within and differentiation among natural populations, assess the relative importance of pollen and seeds as agents of gene flow, and determine whether natural populations experienced a decline in size (bottleneck effect). Does it mean that data of the two cultivated populations were not used, because one cannot consider them as natural populations?
Hardy-Weinberg equilibrium tests were carried out to test for the presence of null alleles, as indicated on line 130. However, it would be interesting to know what the frequency of the null alleles was. Such frequencies could be obtained using the Micro-Checker software, for instance.

I have also the following minor points to share.
L. 216: You have stated: “Adopting a P-value of 0.001, no significant correlation between pairwise genetic distance [FST/(1 − FST)] and geographic distance (in kilometers) was found using the Mantel test …”. What s the justification of the P-value level? Why not P-value < 0.01?
L. 244: Are you referring to the Zhang et al. 2016 paper?
L. 262-264: This sentence is not clear. Please rewrite.
L. 277: How did you determine that r was biased?

Validity of the findings

With regard to the validity of the findings, the results appears to be quite novel for Lindera glauca. However, my my main concern is with the potential confounding effect of small sample size in the observed genetic parameter values, as indicated in the previous section. Although the number of populations as well as the number of sampled individuals are larger than that of a previous paper on the same species (Zhu et al. 2016, if I am not wrong), I cannot rule out the possibility that the lower within population diversity observed in the current study is only due the small number of individuals sampled in many populations. We can see on Table 2, that almost all the populations represented by at least 20 individuals have the highest percentages of polymorphic loci and the largest number of alleles for nuclear markers.

The authors have indicated that they have taken measures to avoid sampling clones but they nevertheless ended up with presence of few clones, and copies did not seem to have been removed for the analyses. This should have been done at least in an additional analysis to evaluate the impact of the presence of clones in the results obtained.

---

## Round 0.8 · accepted · Accept

The authors have to note a clear and frank statement on the limitations of the study and the sample size, also in the Abstract.

·

Basic reporting

I have no additional comments.

Experimental design

I have no additional comments.

Validity of the findings

I have no additional comments.

Additional comments

I still think that the sample size of most of the populations is under the standards for a population genetics study (at least for nuclear markers). Moreover, I still do not see the added-value of including the two cultivated populations in this study. I will let the Editor make his decision, as I do not know the minimum standards required by PeerJ.